# Bundling and segregation affect pheromone deposition, but not choice, in an ant

**Massimo De Agrò[1,2]\*, Chiara Matschunas[1], Tomer J Czaczkes[1]**

[1]Animal Comparative Economics laboratory, Department of Zoology, University of Regensburg, Regensburg, Germany; [2]Department of Biology, University of Florence, Florence, Italy

**Abstract** Behavioural economists have identified many psychological manipulations which affect perceived value. A prominent example of this is bundling, in which several small gains (or costs) are experienced as more valuable (or costly) than if the same total amount is presented together. While extensively demonstrated in humans, to our knowledge this effect has never been investigated in an animal, let alone an invertebrate. We trained individual *Lasius niger* workers to two of three conditions in which either costs (travel distance), gains (sucrose reward), or both were either bundled or segregated: (1) both costs and gains bundled, (2) both segregated, and (3) only gains segregated. We recorded pheromone deposition on the ants' return trips to the nest as measure of perceived value. After training, we offer the ants a binary choice between odours associated with the treatments. While bundling treatment did not affect binary choice, it strongly influenced pheromone deposition. Ants deposited c. 80% more pheromone when rewards were segregated but costs bundled as compared with both costs and rewards being bundled. This pattern is further complicated by the pairwise experience each animal made, and which of the treatments it experiences first during training. This demonstrates that even insects are influenced by bundling effects. We propose that the deviation between binary choice and pheromone deposition in this case may be due to a possible linearity in distance perception in ants, while almost all other sensory perception in animals is logarithmic.

**\*For correspondence:**
massimo.deagro@gmail.com

**Competing interest:** The authors declare that no competing interests exist.

## Editor's evaluation

This innovative study investigates the presence of biases in value perception in ants. The authors were able to show that the distribution of rewards and costs influences perceived reward value in ants, an effect that was observed for pheromone deposition but not choice behaviour.

## Introduction

Broadly speaking, when an animal must choose between options, it can employ one of three different strategies, characterized by different levels of precision: random choice; following a heuristic or rule of thumb; or comparison of outcome value and choice for the highest. Traditional economic theory, exemplified by expected utility theory, assumes (human) decision-makers are rational and perform strict value-based choice (*Mankiw, 2011*). The now well-established field of behavioural economics has vigorously pushed back against this idea, demonstrating that humans often make decisions which are not fully logical, economically rational, or 'optimal' (*Camerer et al., 2011*; *Tversky and Kahneman, 1974*), even when actively comparing options.

A major insight lying at the heart of behavioural economics is that value is *perceived*. This can lead directly to deviations from optimality: between the acquisition of the information and the evaluation of possible outcomes, something gets lost in translation. As has been well established for over a century by the study of pyschophysics, perception is non-linear, usually on a logarithmic scale (*Gescheider, 1997*). Value perception for humans is likewise non-linear, as famously stated by Kahneman and Tversky, and extensively demonstrated thereafter (*Camerer, 2004*; *Kahneman and Tversky, 1979*). Moreover, humans weigh losses more strongly than gains. Finally, value is relative, usually to an expectation or some sort of anchor, see *Figure 1*.

The non-linear nature of perceived value results in many behavioural biases, a very prominent one being the bundling vs. segregation effect. Crucial to the current experiment, the perceived value of a compound item or option can be changed by presenting it either as one option (bundling) or as multiple small parts (segregation). Due to diminishing returns, bundling results in a weaker total sensation than segregation – either lower value for positive value options or lower cost for negative value options (see *Figure 1*). This is because more of the sensation occurs on the shallow part of the curve. This fact is regularly exploited by consumer psychologists and marketing experts, for example by bundling option when selling new cars: when spending €50,000 on a new car, spending €51,000 for the model with included sound system might not be experienced as painfully costly, even if, considered by itself, €1000 might be more than most people are willing to spend on a sound system for a car. Bundling and segregation have been extensively studied by consumer psychologists (*Johnson et al., 1999*; *Naylor and Frank, 2001*; *Noone and Mattila, 2009*).

As in the study of human economics, non-human animals have been treated and modelled as rational economic agents, leading to deep insights into animal behaviour via the optimal foraging theory framework (*Davies, 2012*; *Emlen, 1966*; *MacArthur and Pianka, 1966*; *Pyke et al., 1977*). However, often inspired directly by behavioural economic research on humans, predictable deviations from optimality and rationality have been described (*Zentall, 2015*). For example, pigeons, rats, and ants all show a preference for high-effort over low-effort associated options (*Clement et al., 2000*; *Czaczkes et al., 2018a*; *Lydall et al., 2010*), much as humans do (*Norton et al., 2012*). Similarly, much as in humans, the addition of an irrelevant option in an option set (a 'decoy') can change the preference structure in many animals, including birds, cats, bees, and ants (*Bateson et al., 2002*; *Bateson et al., 2003*; *Parrish et al., 2015a*; *Sasaki and Pratt, 2011*; *Scarpi, 2011*; *Schuck-Paim et al., 2004*; *Shafir et al., 2002*).

Ants and bees also show relative value perception, changing the perceived value of an option depending on their expectations (*Bitterman, 1976*; *Couvillon and Bitterman, 1984*; *Wendt et al., 2019*; *Wendt and Czaczkes, 2020*). An ant which is trained to expect a very sweet reward, for example, will be more likely to reject a moderate reward than an ant which was expecting a moderate one – a negative contrast. Likewise, ants expecting a very mild reward are more likely to accept a moderate reward than ants which were expecting that quality – a positive contrast (*Wendt et al., 2019*). These changes are also mirrored in the ants' deposition of recruitment pheromone: ants deposit more pheromone to resources they perceive as higher quality (*Beckers et al., 1993*; *Jackson and Châline, 2007*), and indeed ants deposit more pheromone for moderate rewards if they had been expected poor quality, and less pheromone if they were expecting high quality. This demonstrates that a key aspect of prospect theory – relative value perception – is present in insects as well as in humans.

Crucially for the current experiment, there is evidence that social insects, especially ants, perceive value logarithmically. This is perhaps not surprising, given than logarithmic perception is a key assumption of the well-established Weber-Fechner law (*Fechner, 1860*; *Weber, 1834*), which describes the psychophysics of perception. *Wendt et al., 2019*, demonstrated a faster rise in food acceptance in the lower range of acceptance food qualities (e.g. from 0.1 to 0.3 molar) than in the higher range (e.g. from 0.5 to 1.5 molar). Recently, *De Agrò et al., 2021*, demonstrated that *Lasius niger* ants have a strong aversion to risky food sources (i.e. with fluctuating quality), which could be fully explained by logarithmic value perception, as predicted by prospect theory. Ants prefer a certain food source offering 0.55 M sucrose to one which fluctuates between 0.1 and 1.0, but if the options are logarithmically balanced (0.3 M vs. 0.1 or 0.9), ants are completely indifferent.

While the bundling vs. segregation effect is extremely well studied in humans, surprisingly, to our knowledge no attempt has been made to examine it in animals. In a related study, chimps were shown to prefer whole rewards (potato chips) to rewards which were broken into smaller pieces, even when

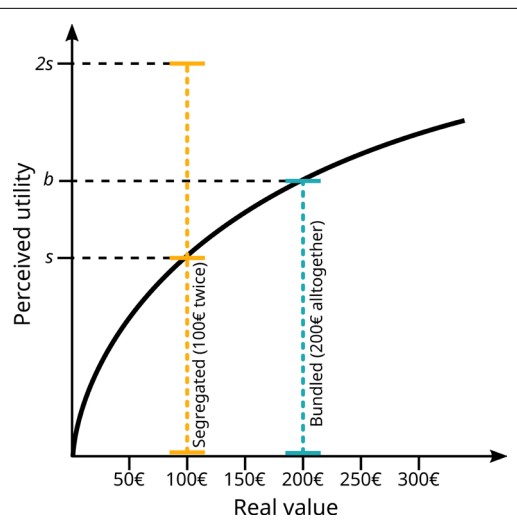

**Figure 1.** Simplified schematic of prospect theory from **Kahneman and Tversky, 1979**, with a graphical illustration of bundling and segregation. On the x-axis, actual value of a gain or a loss (here exemplified with money). Perceived utility does not scale linearly with value, but logarithmically. Receiving a gain of €100 (segregated) twice will produce a level of 'happiness' of '2s', more than the level 'b' perceived when receiving €200 all together (bundled). The same is true for 'losing' the same amounts, where two losses of €100 are felt stronger than a single one of €200.

the broken rewards had an higher absolute quantity (**Parrish et al., 2015b**). However, consumption of the whole and broken rewards took the same amount of time, and rewards were chosen before consumption could begin, so it is likely that both broken and whole rewards were considered as one unitary bundle.

In order to study bundling and segregation in ants, we segregated rewards spatially, using small sucrose drops along a linear runway. However, while this segregates rewards, it also segregates a potential cost – the walking distance to the reward. Thus, we first demonstrate that ants do indeed prefer closer to farther food sources, and thus that food distance is considered a cost. We then train individual ants to two of three treatments: rewards and costs bundled ('bundled'), rewards and costs segregated ('segregated all'), and rewards segregated but costs bundled ('segregated reward'). We record pheromone depositions on the ant's return from each treatment type, and then ask ants to choose between the pair of treatments they were trained on. We predicted that the 'segregated reward' treatment would be the preferred option, as bundling of costs should minimize their impact, and segregation of rewards should boost theirs. As we did not know the relative strength of the rewards and costs, we had no strong a priori predictions about the relative perception of 'bundled' and 'segregated all'. However, as prospect theory predicts that losses are weighted more strongly than gains (**Kahneman and Tversky, 1979**), we had a weak expectation that 'bundled' would be perceived as slightly better than 'all segregated'.

## Materials and methods
### Subjects

We used a total of 144 *L. niger* individuals, coming from 19 queen-less colony fragments, consisting of around 1000 ants each. Sample size was set a priori, based on the time and resources available (**Lakens, 2022**), while maintaining a minimum amount of ants per condition based on our previous experience with these types of experiment. Each fragment was collected from a different wild colony on the University of Regensburg campus. Workers from colony fragments forage, deposit pheromone, and learn well (**Evison et al., 2008**; **Oberhauser et al., 2018**). Each fragment was housed in a transparent plastic box (30 × 20 × 40 cm³), with a layer of plaster on the bottom. A circular plaster nest, 14 cm in diameter and 2 cm thick, was also provided. The colonies were kept at room temperature (21–25°C) and humidity (45–55%), on 12:12 light:dark cycle for around 9 months. Each colony was fed exclusively on 0.5 M sucrose solution ad libitum, and deprived of food 4 days prior to each test. Water was provided ad libitum and was always present.

### Procedure

All four experiments reported in this paper used a conditioning procedure described in **Czaczkes, 2018c**. The procedure was generally the same, with a few modifications dictated by the specific conditions.

For each tested subject the procedure started by connecting a drawbridge to the nest box. This bridge was composed of a 20 cm long, 1 cm wide, slanted section, one end of which laid on the

plaster floor of the nest box. The other end led to a straight 10 cm long, 1 cm wide, runway section. Both of these were covered by unscented paper overlays. Depending on the visit, this bridge could lead either to a straight runway or to the stem of a Y-maze. In the first visit, multiple ants were allowed on the bridge. A 0.5 M sucrose solution drop was placed at the end of the bridge. The first ant to reach the drop and start drinking was marked on the abdomen with a dot of acrylic paint. The non-marked ants were gently returned to the nest box, while the marked one was allowed to drink to satiation, then allowed to return to the nest on her own. In the nest the marked ant performed trophallaxis (mouth to mouth food transfer) with nest-mates, and then returned to the bridge location ready for the following visit, which varied depending on the experiment being run.

## Pilot experiment – Is increased food distance negatively perceived?

As described in the Introduction, segregated rewards should be perceived as being of higher value than an equal-quality bundled alternative. However, this is also true for punishments. Since in our experiment we segregated rewards by placing them on different parts of a long runway (see next paragraph), we needed to test first whether increased travelled distance makes rewards less preferred.

The previously marked ant was allowed onto the bridge. This time, the bridge was attached to either a 25 or 75 cm long runway (systematically varied). This runway was covered with a scented paper overlay (either rose or lemon odour). Scenting was achieved by storing the overlays in a sealed plastic box with three drops of food flavouring for at least 24 hr. At the end of the runway, we placed a high-quality (1.5 M) sucrose solution drop, flavoured with the same smell as the runway (at a ratio of 1 µl food flavour per ml sucrose solution). The ant eventually found the drop, drank to satiation, and then went back to the nest. At this point, we discarded the scented overlays, in order to remove the deposited pheromone.

As soon as the ant unloaded, it was allowed back onto the bridge again. The bridge now connected to the reciprocal runway length (25 cm if the previous visit was to 75, and vice versa). This runway was covered with paper scented with a different smell to the previous visit. At the end of the runway, the ant again found a drop of 1.5 M sucrose solution, again flavoured to match the paper overlay. After drinking, the ant again allowed to return to the nest.

The same procedure was repeated another time. Thus, the ant experienced the long and the short runways twice each. For all visits, we measured the number of times the ant deposited pheromone on the scented runway, both the way towards the drop and the way back. Pheromone deposition in *L. niger* is a stereotyped behaviour, in which an ant pauses for c. 0.2 s, and curls its abdomen down, pressing it firmly onto the substrate (*Figure 2A*, *Video 1*). This behaviour is easily quantified by eye (*Beckers et al., 1993*). Pheromone deposition co-varies with the (perceived) value of a resource, with ants making more depositions for resources they consider to be high quality (*Beckers et al., 1993*; *Wendt et al., 2019*; *Wendt and Czaczkes, 2020*).

After these four visits, the ant was allowed onto the bridge one last time. This time, the bridge was connected to the 10 cm long stem of a Y-maze (*Figure 2B*). The stem was covered with unscented paper, and tapered to a 2 mm wide point. Here, the two arms of the Y-maze started, also tapered, in order to ensure that the ant to contact both arms at the same time once at the end of the stem. One of the two arms was scented with the long runway odour, while the other was scented with the short runway odour. We noted on which of the two arms the ant ran for at least 2 cm (considered the 'initial' decision), and at the end of which of the two it arrived first (considered the 'final' decision). Once this happened, the ant was picked up with a piece of paper and moved back at the start of the stem to be retested. This way, we could test the ants' preference three times. After the test, the ant was permanently removed from the colony.

A total of 24 ants from five different colonies was used for this experiment.

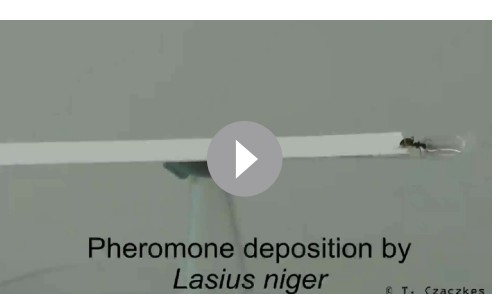

**Video 1.** Pheromone deposition example.
https://elifesciences.org/articles/79314/figures#video1

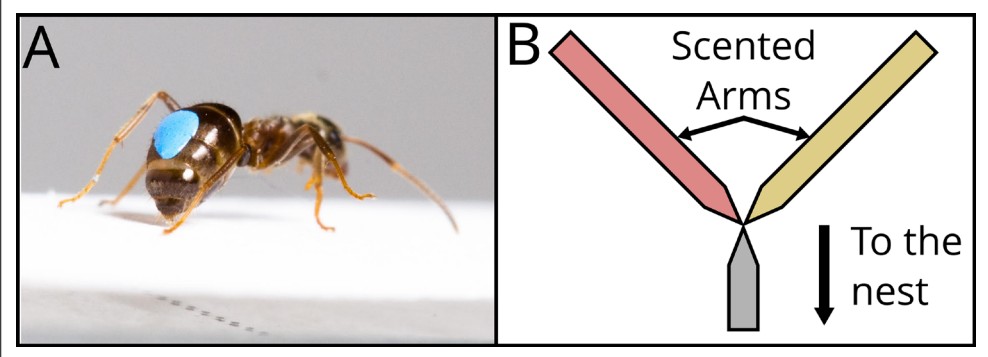

**Figure 2.** Description of the two dependent variables recorded. In (**A**), a marked *Lasius niger* ant pauses and presses its abdomen to the runway, leaving pheromone mark. See *Video 1*. The number of pheromone depositions was counted. Photo : Julia Giehr. (**B**) shows a schematic representation of the Y-maze used for the binary choice test. Coming from the nest, the ant walks on an unscented runway, constituting the Y-maze stem, until it reaches the bifurcation. The two Y-maze arms were scented with two different odours, corresponding to the ones present during the previous visits and associated with the two experienced treatments. The bifurcation tapers in the middle to ensure that the ant senses both odours before making a choice.

## Main experiment – bundling vs. segregation

Having established that increased travel distance makes food sources less attractive (see Results), we proceeded with the main experiment.

The procedure is very similar to the pilot. Each marked ant was allowed onto the bridge, after which it was presented with a scented runway. This scented runway could correspond to one of three different treatments:

### 'Segregated all' (rewards and costs) (*Figure 3A*)

The ant encountered a 75 cm long runway. Every 25 cm, the runway tapered to 2 mm in width with a 0.2 µl, 1.5 M, sucrose solution drop on the taper. After the drop, the runway widened again, until the next 25 cm taper. Here, the ant found a second drop, identical to the previously encountered one. The runway proceeded for a third 25 cm section, ending in a large drop. To avoid evaporation, as well as limiting the risk of the ant bypassing the rewards without noticing, the drops were delivered with a micropipette when the ant reached the designated position, rather than being placed prior to testing.

In this treatment, the reward (drops) was segregated into three different experiences, and as such its combined value should be perceived as higher than one being presented as a single one. The volume of the first two drops was selected in order to ensure that the ant would reach the third drop without becoming satiated after the first or the second one: The crop volume of *L. niger* foragers is under 1 µl (*Mailleux et al., 2000*), and ants which encounter such drops drink them and then continue walking forwards (*Czaczkes et al., 2019*). The last drop instead was much larger, to ensure that ants could drink to satiation, and thus avoiding other possible discounting effects, such as disliking not being completely satiated (*Mailleux et al., 2006*; *Mailleux et al., 2005*).

However, in this treatment also the cost is segregated: Rather than being experienced as a single 75 cm long runway, the ant encountered three 25 cm ones, interspersed by rewards. Thus, this condition is expected to enhance both the perceived value and the perceived cost.

### 'Segregated reward' (bundled costs) (*Figure 3B*)

In this second treatment, the ant encountered a 75 cm long runway. At every 25 cm mark, a narrowing portion was present, but in this treatment no small sugar drops were provided. The narrowing of the paper was maintained to ensure consistency with the previous treatment. To assure consistency among treatments, the experimenter followed the same procedure of the 'segregated all condition': the micropipette was brought to the narrowing point at the end of the 25 cm runway when the ant got near it and the plunger depressed delivering no drop (i.e. a sham treatment). At the end of the 75 cm, two 0.2 µl, 1.5 M drops were presented in short succession, just 5 mm from each other. After

another 5 mm, the third ad libitum drop was placed. Thus, in this treatment, the distance travelled (=cost) remained bundled, while the reward was still segregated.

### 'Bundled' (rewards and costs) (*Figure 3C*)

In this third treatment, the ants encountered the same runway as above. However, instead of presenting three drops at the end, only the third ad libitum drop was offered. Thus, in this treatment, both the reward and the cost are bundled. The sham pipetting was also carried out in this treatment.

## Pairwise training and a priori hypotheses

We performed three different conditions, corresponding to the three different pairings of the three treatments (A vs. B, A vs. C, B vs. C). Forty ants were tested per condition, for a total of 120. Each ant would experience one of the three treatments for the first visit, associated with a distinct odour. On the subsequent one, the animal encountered a second treatment, associated with another odour. This procedure was repeated for four times, for a total of eight visits alternating between the two selected treatments. This way, each ant experienced four times each treatment. On the way back to the nest, we counted the pheromone deposited by each ant (*Figure 2A*) in each of the three runway sections. In the end, the ant was presented with a Y-maze, and had to choose between the two treatment odours. As for the pilot experiment, the Y-maze test (*Figure 2B*) was repeated three times, by picking up the ant immediately upon reaching the end of the chosen arm and placing it back at the start of the stem.

Broadly, we expect the following preference structure:

> In condition 1 (B vs. C), the 'segregated reward' treatment should be preferred over the 'bundled' one. This is expected due to the introduced bundling effect.
> In condition 2 (B vs. A), the 'segregated reward' treatment should be preferred over the 'segregated all' one. This is expected as the boosting of reward by segregation effect is identical, while the 'segregated all' condition also boost the cost.
> In condition 3 (A vs. C), the 'bundled' treatment should be preferred over the 'segregated all' one. This is because costs are often more heavily weighted than gains (*Lakshminarayanan et al., 2011*; *Tversky and Kahneman, 1981*), and so boosting both rewards and costs by the same factor will tend to emphasize the costs more. Note that this was a tentative, and weak, expectation, as we had no a priori way of knowing how the cost of walking a set distance compares to a set sucrose reward.

## Data analysis

The entire statistical analysis code, including data handling, figure code, and analysis results, is presented in supplement ESM2. Raw data is available in supplement ESM1. All the statistical analyses were performed in R 4.1.2 (*R Development Core Team, 2020*). The packages readODS (*Schutten et al., 2020*) and reshape2 (*Wickham, 2007*) were used to load and prepare the data. We focused on two measures: the binomial choice at the last experimental visit, and the number of pheromone depositions during the training visits for the different options.

To analyse the former, we employed generalized mixed effect models with a binomial distribution using the package glmmTMB (*Brooks et al., 2017*; *Magnusson et al., 2020*). In every experiment, we included as predictors the choice order (first, second, or third visit to the Y-maze) and decision line (initial decision, passed the first 2 cm line; or final decision, reached the end of the arm). The ant identity nested in the colony of origin was included as a random effect. The goodness of fit was evaluated with the package 'DHARMa' (*Hartig, 2018*). We performed an analysis of deviance to observe the effect of the predictors using the package car (*Fox and Weisberg, 2011*), and then performed Bonferroni-corrected post hoc analysis on predictors that have an effect using the package emmeans (*Lenth et al., 2020*). Lastly, we computed effect sizes (*Ben-Shachar et al., 2020*).

For pheromone deposition we followed the same procedure. We employed GLMM with a Poisson error structure, varied into a Tweedie error structure when DHARMa testing suggested that as appropriate. In the pilot, we included direction (to the drop or back to the nest) and visit length (short or long) as predictors. For the other three conditions, we included runway section (nearest to the nest, middle, nearest to the end) and treatment of the visit ('bundled', 'segregated all', 'segregated reward'). The ant identity nested in the colony of origin was included as a random intercept, with

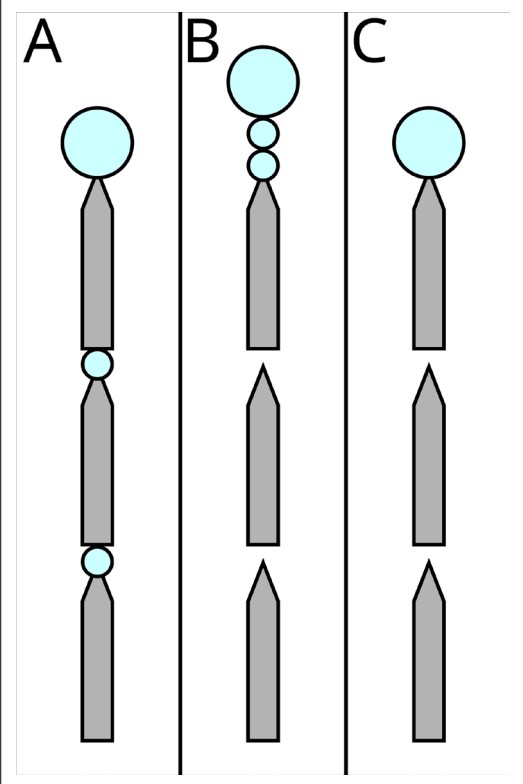

**Figure 3.** The three possible experimental treatments. Grey shapes represent the runway segments, each 25 cm long, 1 cm wide, tapering to 2 mm to ensure that ants encounter the sucrose drops. Big blue circles represent ad libitum 1.5 M sucrose solution, small circles represent 0.2 µl drops which the ants can drink, but will not satiate them. In (**A**) 'segregated all', both the costs (travel over the runways) and the rewards (drops of 1.5 M sucrose, blue circles) are segregated. In (**B**) 'segregated rewards', only the rewards are segregated. In (**C**) 'bundled' both the costs and the rewards are bundled.

repeated visits as a random slope. The random effect was simplified by removing the repeated visits in models that failed to converge. We then followed up with analyses of deviance and post hoc analyses as well.

In the pilot experiment, we expected the ants to deposit more pheromone on the long runway, independently of their preference. All things being equal, a triple-length runway will offer triple the pheromone deposition time. To control for this bias, we multiplied the observed pheromone deposited on the short runway by three. However, *L. niger* are reported to deposit more pheromone nearer to the food source (*Beckers et al., 1992*). As such, a simple multiplication may have still not be fully appropriate as a comparison. The pheromone deposition for the pilot experiment is reported for completeness, but we thus advise caution in the interpretation of this data. This is not a problem for the three experimental conditions as the runway length remains fixed.

After analysis, the data was then passed onto a Python 3 (*Van Rossum and Drake, 2009*) environment using the package reticulate (*Ushey et al., 2021*), to produce graphs. To achieve this, we used the libraries pandas (*Reback et al., 2020*), numpy (*Oliphant, 2006*; *van der Walt et al., 2011*), matplotlib (*Hunter, 2007*), and seaborn (*Waskom et al., 2017*).

Examining the results of the aforementioned model, we discovered a possible contrast effect, as depending on the condition, the amount of pheromone deposited for the same treatment changed abruptly. It is indeed crucial to consider how the first experience of ants can have ripple effect on its subsequent decisions (*De Agrò et al., 2021*). To further examine this effect we remodelled the data including the first experienced treatment as a factor. We will present the results of the model that included and that did not include the first experienced treatment as a factor separately. The full analysis, and all the raw data, can be found in the supplements.

## Results

### Pilot experiment – cost of travelled distance

In the pilot experiment, the ants were asked to choose between two odours: one associated with a short runway and the other associated with a long one. We observed a 90% probability of the ants choosing the short-associated odour when encountering the Y-maze for the first time, significantly higher than chance level (GLMM post hoc: prob.=0.896, SE = 0.056, DF = 136, t=3.589, p=0.0014). The probability quickly dropped to chance level for the two subsequent visits (visit 2: prob.=0.547, SE = 0.124, DF = 136, t=0.376, p=1; visit 3: prob.=0.484, SE = 0.123, DF = 136, t=−0.129, p=1). This is generally to be expected in this type of experiment, as the lack of a reward and manipulation easily disrupts the ant decision.

The pheromone deposition analysis confirms this pattern, as indeed ants deposit double the amount of pheromone per unit length on the short runway over the long one (GLMM post hoc ratio = 2, SE = 0.231, DF = 185, t=5.973, p<0.0001).

## Main experiment – bundling vs. segregation

### Condition 1: 'Segregated reward' vs. 'bundled'

In this experiment, we expected ants to prefer the 'segregated reward' treatment over the 'bundled' one due to bundling.

We observed an odd difference between the three subsequent tests on the Y-maze (GLMM ANODA, chi-square=9.5744, DF = 2, p=0.0083). Specifically, the ants showed no significant preference in the first visit (GLMM post hoc: prob.=0.516, SE = 0.0693, DF = 232, t=0.227, p=1) nor in the third (prob.=0.486, SE = 0.0693, DF = 232, t=−0.195, p=1). However, they significantly preferred the segregated option in the second visit (prob.=0.729, SE = 0.0596, DF = 232, t=3.279, p=0.0036). We consider this a false positive (see Discussion).

For the pheromone deposition (*Figure 4*), we observed no difference between the treatments (GLMM ANODA, chi-square=2.0487, DF = 1, p=0.1523). Here, we also observed a difference between the three sections (chi-square=107.2664, DF = 2, p<0.0001), and there was no effect of the interaction between the two other predictors (chi-square=0.6412, DF = 2, p=0.73). Specifically, there was no difference in the pheromone deposited for the 'segregated reward' option than for the 'bundled' one (GLMM post hoc: ratio = 0.71, SE = 0.1735, DF = 948, t=−1.399, p=0.6482). The ants deposited overall more pheromone on the section of the runway nearest the drop in respect to the second (ratio = 1.55, SE = 0.0955, DF = 948, t=7.124, p<0.0001) and in the second in respect to the third (ratio = 1.2, SE = 0.0842, DF = 948, t=2.599, p=0.038) section.

Regarding the effect of the first encountered treatment (*Figure 5*), we observed an effect of the first encountered treatment (GLMM ANODA, chi-square=5.8514, DF = 1, p=0.01556). The effect disappears when looking at the post hoc (GLMM post hoc: ratio = 0.542, SE = 0.139, DF = 830, t=−2.382, p=0.1048), probably due to the weight of Bonferroni correction. Even if not significant, it seems that ants deposited slightly more pheromone for the segregated reward option when it was encountered first in respect to when it was encountered second (ratio = 0.354, SE = 0.141, DF = 830, t=−2.613, p=0.0549).

### Condition 2: 'Segregated reward' vs. 'segregated all'

In this experiment we expected the 'segregated reward' treatment to be preferred over the 'segregated all' treatment.

We found no difference between subsequent Y-maze visits (GLMM ANODA: chi-square=3.1754, DF = 2, p=0.2044), and we observed no overall significant preference in the Y-maze test (GLMM post hoc: prob.=0.524, SE = 0.06, DF = 232, t=0.402, p=0.688).

Regarding pheromone deposition (*Figure 4*), we observed a difference between the treatments (GLMM ANODA, chi-square=60.7675, DF = 1, p<0.0001) and a difference between the three sections (chi-square=10.4118, DF = 2, p=0.0055). We did not observe a statistically significant effect of the interaction (chi-square=5.667, DF = 2, p=0.0588). Specifically, the ants deposited more pheromone for the 'segregated reward' option compared to the segregated cost one (GLMM post hoc: ratio = 0.555, SE = 0.0422, DF = 945, t=−7.746, p<0.0001). In the 'segregated reward' visits, but not the 'segregated all' visits, the ants deposited more pheromone on the section of the runway nearest the drop relative to the furthest one (ratio = 1.58, SE = 0.1809, DF = 945, t=3.995, p=0.0007).

Regarding the effect of the first encountered treatment (*Figure 5*), the ants deposited overall more pheromone when they encountered the 'segregated reward' option first, irrespective of the currently experienced treatment (GLMM post hoc: ratio = 0.472, SE = 0.0991, DF = 830, t=−3.576, p=0.002). We also observed that when the 'segregated reward' treatment was encountered first, the ant significantly preferred it to the 'segregated all' option (ratio = 0.296, SE = 0.1081, DF = 830, t=−3.333, p=0.0054) section. However, when they encountered the 'segregated all' treatment first the ants showed no difference (ratio = 0.703, SE = 0.2536, DF = 830, t=−0.976, p=1). Ants deposited more pheromone for the 'segregated reward' option when it was encountered first in respect to when it was encountered second (ratio = 0.306, SE = 0.1092, DF = 830, t=−3.318, p=0.0057).

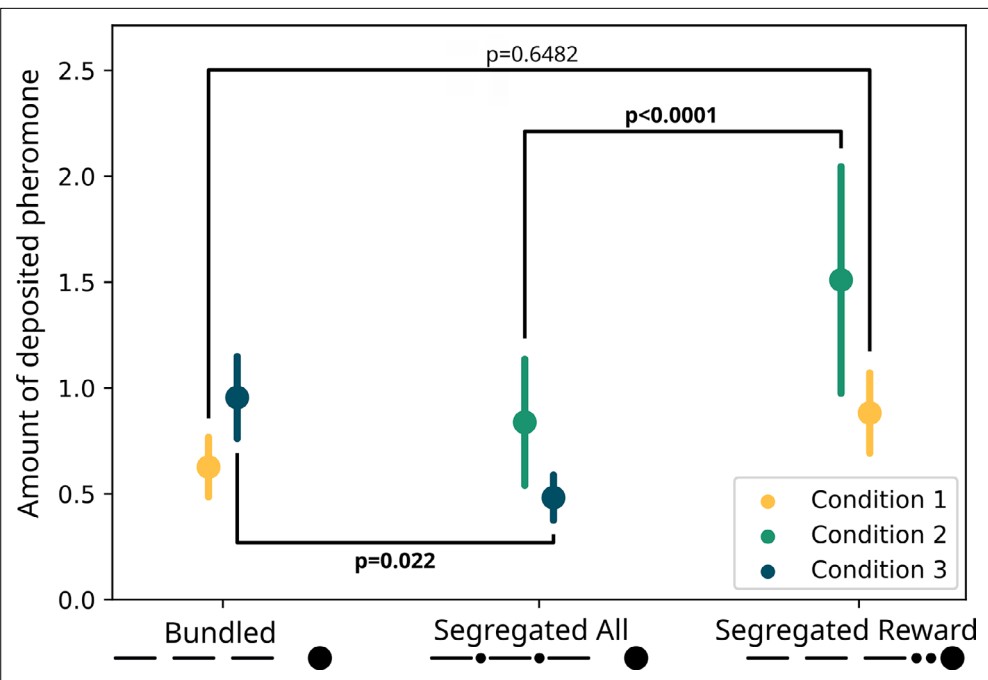

**Figure 4.** Modelled pheromone deposition for each treatment, across the three condition. Y-axis: amount of deposited pheromone per runway section. Error bars represent standard error. In yellow, pheromone deposited in the 'bundled' vs. 'segregated reward' condition (n=40). The two treatments are not significantly different from each other (glmm post-hoc p=0.6482). In green, pheromone deposited in the 'segregated all' vs. 'segregated reward' condition (n=40). The two treatments are significantly different from each other (glmm post-hoc p<0.0001). In blue, pheromone deposited in the 'segregated all' vs. 'bundled' condition (n=40). The two treatments are significantly different from each other (glmm post-hoc p=0.022). All p-values are corrected for multiple testing.

## Condition 3: 'Segregated all' vs. 'bundled'

In this experiment, 'bundled' treatment was expected to be preferred over the 'segregated all' one.

The ants showed no overall significant preference for either treatment (GLMM post hoc prob.=0.466, SE = 0.122, DF = 232, t=−0.279, p=0.7803).

However, we saw large differences in pheromone deposition between the treatments (*Figure 4*). We found an effect of the two treatments (GLMM ANODA, chi-square=9.9822, DF = 1, p=0.00158), of the three runway sections (chi-square=24.606, DF = 2, p<0.0001) and of the interaction between those (chi-square=13.3183, DF = 2, p=0.00128). Specifically, ants deposited more pheromone in the 'bundled' visits than in the segregated cost ones (GLMM post hoc: ratio = 0.0505, SE = 0.1124, DF = 948, t=−3.07, p<0.0001). Moreover, in the 'bundled' visits, they deposited more pheromone on the runway section nearest the drop relative to the the middle section (ratio = 1.534, SE = 0.1427, t=4.6, p<0.0001) or the section nearest the bridge (ratio = 1.701, SE = 0.1626, DF = 948, t=5.557, p<0.0001). This pattern was not present in the segregated cost visits (first vs. second section: ratio = 1.055, SE = 0.1229, DF = 948, t=0.459, p=1; first vs. third section: ratio = 1.007, SE = 0.1161, DF = 948, t=−0.06, p=1).

Regarding the effect of the first encountered treatment (*Figure 5*), the ants deposited overall more pheromone when they encountered the 'segregated all' option first, irrespective of the currently experienced treatment (GLMM post hoc: ratio = 2.909, SE = 0.643, DF = 830, t=4.832, p<0.0001). We observed that when the 'bundled' treatment was encountered first, the ant significantly preferred it to the 'segregated all' option (ratio = 0.322, SE = 0.120, DF = 830, t=−3.032, p=0.015) section. Instead, when they encountered the 'segregated all' treatment first the ants showed no difference (ratio = 0.916, SE = 0.305 t=−0.263, p=1). Ants deposited more pheromone for the 'segregated all' option when it was encountered first in respect to when it was encountered second (ratio = 4.905, SE = 1.787, DF = 830, t=4.365, p=0.0001).

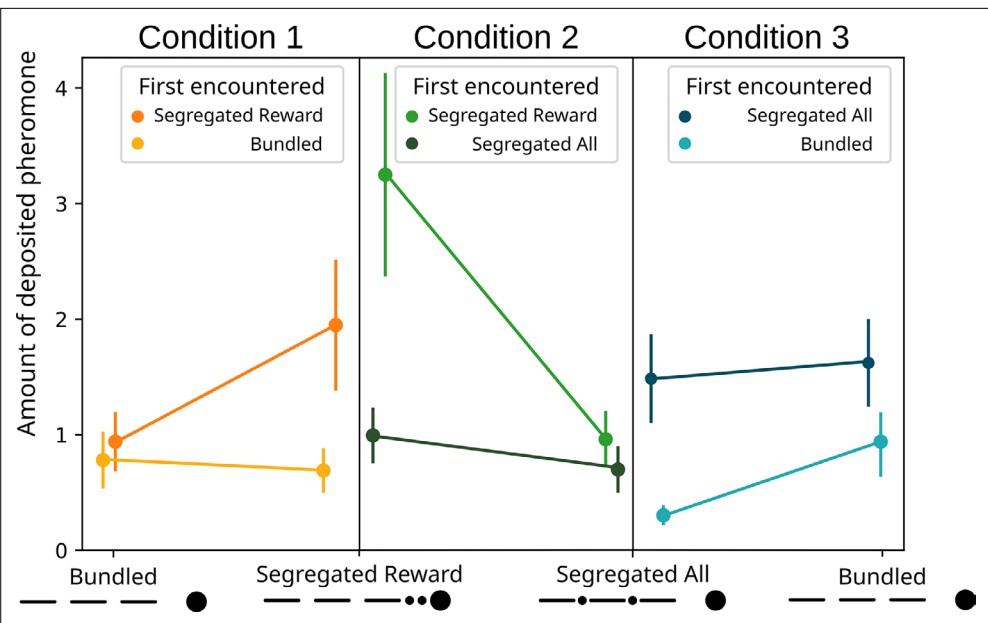

**Figure 5.** Modelled pheromone deposition for each treatment, across the three condition, including the influence of the first experienced option. Y-axis: amount of deposited pheromone per runway section. Error bars represent standard error. n=40 for each condition. When 'segregated reward' is encountered first, in condition 2, the overall pheromone deposited is higher in respect to the 'segregated all' option (glmm post-hoc p=0.0022). In condition 3, when the 'bundled' option is encountered first, the overall pheromone deposited is lower in respect to the 'segregated all' option (glmm post-hoc p=0.0001).

## Discussion

Bundling and segregation treatment strongly affected the ants' pheromone deposition, but not their choices. No consistent preference was found in the binary choices of the ants from the main study. While ants significantly preferred the segregated reward over the bundled reward in their second of three trials, we can conceive of no plausible biological or psychological reason for this to be so, and interpret this as a false positive. Thus, there is no evidence that bundling or segregation affect choice.

However, the pattern of pheromone deposition, which correlates strongly with perceived food quality in these and other ants (*Beckers et al., 1993*; *Czaczkes et al., 2018b*; *Jackson and Châline, 2007*; *Wendt et al., 2019*), was partially in line with our predictions: ants deposited most pheromone when the costs were bundled and rewards segregated, and less in the other two treatments (*Figure 4*). Our weaker prediction of 'bundled' presenting a higher deposition than 'segregated all' was also supported. Thus, pheromone deposition seems to be in line with predictions from classical behavioural economic and perception research (*Camerer, 2004*; *Johnson et al., 1999*; *Kahneman and Tversky, 1979*; *Naylor and Frank, 2001*; *Noone and Mattila, 2009*). Specifically, we observed more deposition for 'bundled' over 'segregated all', and for 'segregated reward' over 'segregated all'. By contrast, the preference for 'segregated reward' over 'bundled' is not significant. This seems counter-intuitive, as the 'bundled' and 'segregated reward' options only differ in the boosting of a gain. Moreover, the contrast of 'segregated all' vs. 'bundled' – for which we had no strong a priori prediction as we had no idea how gains and losses are weighted – shows a very clear difference.

The observed result suggests two, non-mutually exclusive, interpretations: Either ants experience a strong segregation effect for losses, and a very mild one, if any, for gains; or our treatment failed to segregate rewards effectively. If the first interpretation is true, the 'bundled' and the 'segregated reward' treatments will be perceived as almost identical. In the contrast between 'segregated all' and 'bundled', the higher number of depositions for bundled makes sense, as the segregation of losses drives the preference completely. The higher deposition for 'segregated reward' over 'segregated all' remains equally clear, as the two options only differ in the realm of losses. If our second interpretation is correct, the three sucrose solution drops, especially in the 'segregated reward' condition, are

still considered by the animal as a single (i.e. bundled) reward. In the 'segregated all' condition the separation between rewards seems more believable, however the results observed strongly suggest that the reward segregation have a very low effect on the perceived value, being overridden by the segregation of losses. It is impossible to disentangle these two options from the current data.

However, the pattern is somewhat more complicated, as we saw large differences in pheromone deposition for the same treatment, depending on the treatments they were paired with. For example, while ants encountering the 'segregated rewards' deposit twice as much pheromone as 'segregated all' when they are paired, for 'segregated rewards' they deposit much less than when paired with the 'bundled' treatment (see *Figure 4*). A similar pattern is seen with 'bundled', where more pheromone is deposited when paired with 'segregated all' and less when paired with 'segregated reward'. One possibility is that these differences arise due to a series of contrast effects. This pattern is further complicated by a strong effect of first encountered treatment (see *Figure 5*). In every condition the effects strengthen when the preferred option is presented first, and drops to chance level when presented second. This suggest that the bias generated through the bundling process has a similar strength to the bias for the first experienced odour (*Oberhauser, 2019*), and thus they appear to counterbalance each other.

Bundling and segregation of options to modify perceived value in insects may have ecological implications, especially in plant-pollinator interactions. If, by segregation, the perceived value of a reward can be increased, plants may be selected to split rewards amongst multiple smaller flowers in an inflorescence, or flowers on a plant. Similarly, they may be selected to attempt to bundle costs, favouring multi-flower inflorescences, allowing insects to walk between flowers, over multiple small, separate flowers.

The finding that bundling and segregation affect ant value perception adds this behavioural economic effect to several others which have also been shown to affect perceived value in insects, including decoy effects (*Sasaki and Pratt, 2011*; *Shafir et al., 2002*; *Tan et al., 2015*), invested effort increasing perceived value (*Czaczkes et al., 2018a*), relative value perception (*Bitterman, 1976*; *Couvillon and Bitterman, 1984*; *Wendt et al., 2019*), and labelling effects (*Hemingway and Muth, 2022*; *Wendt and Czaczkes, 2020*). It is becoming clear that the underlying patterns driving value perception in insects are in many ways parallel to those of humans. This implies either an extremely early evolutionary origin of shared perceptual mechanisms resulting in shared psychophysical laws or convergent evolution on similarly effective systems. The finding of a bundling and segregation effect, alongside the way in which insects respond to differences between experienced and expected rewards, strongly implies that insects share the same broad value function shape as humans (see *Figure 1*). Thus, we would predict that any other behavioural economic effects in humans, which arise from this value function, to also be present in insects. Still missing is a demonstration of an inflection point – that is, like in humans, losses loom larger than gains for insects. That our results imply bundling to more strongly influence losses than gain (see above) suggests this is the case, but formal testing will be required.

The absence of a significant preference in the Y-maze test, contrasted with the clear effect appreciable in the pheromone deposition, is puzzling. We have to consider the possibility that processes often assumed to be tightly linked to each other are instead separate. In this specific case, choice in the Y-maze depends on memory formation, as the choice is made by mentally comparing two options previously experienced. Pheromone deposition instead happens immediately after experience, and does not require any mental comparison to be expressed.

It is possible that ants have simply failed to associate each odour with the reward treatment, and thus chooses randomly in the Y-maze. This seems highly unlikely, as many examples of similar experiments on the same species are available in the literature, where ants are demonstrated to be extremely capable learners, learning even complex multimodal associations with fewer exposures than used in the current study (*Czaczkes and Kumar, 2020*; *De Agrò et al., 2020*).

The most likely explanation to us is that while a difference in the value of options is perceived, the same difference is not recorded during memory formation. Scalar utility theory (*Kacelnik and Brito e Abreu, 1998*; *Rosenström et al., 2016*) is an influential framework that has been developed to describe decision making under uncertainty. This theory postulates that encoding neurons, responsible for the internal representation of values (i.e. quantity, quality, delay, etc.), have logarithmically spaced sensitivities and specificities. Prospect theory (see Introduction) also assumes a process based

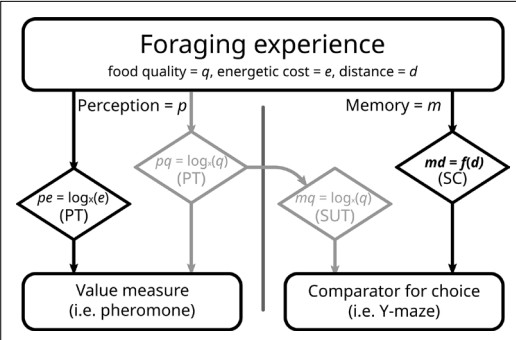

**Figure 6.** Proposed model of perception and memory in light of economic theories. Perception can be the same as the formed memory, but not necessarily. During a foraging bout, the ant perceives (**p**) gains (food quality, **q**) and losses (energy spent to reach, **e**), according to prospect theory (PT). Food quality seems to be registered into memory (**m**) in to the same scale, congruently with scalar utility theory (SUT). Distance travelled (**d**) represents a special case, as it requires precise memory in the context of ant navigation. As such, it may possess a dedicated, direct, and linear memorization circuit (md), like the step counter (SC). In our experiment, we failed to imprint a segregation effect into rewards greyed out boxes; see Results and Discussion, and as such all our options were equal in this realm. With costs perceived logarithmically, but memorized linearly, we would expect the results observed in this experiment.

on a logarithmic function. Thus, both make very similar behavioural predictions, even though one concentrates on value perception, and the other on memory acquisition. However, they do not have to be identical. Most experiments recording both pheromone deposition and subsequent choice use food quality as the measured unit (*De Agrò et al., 2021*; *Oberhauser and Czaczkes, 2018*; *Wendt et al., 2019*). Logarithmic perception of quality is probably the reason a mismatch between choice and instantaneous value perception has eluded discovery. We believe that the mismatch produced in this experiment has to do with the type of cost chosen as treatment: distance travelled. As previously mentioned, our treatment seemed unable to produce a segregation effect for the reward, while showing a strong segregation effect for losses, that is, the travelled distance. It is reasonable to assume that ants perceive value logarithmically, in accordance with prospect theory (*De Agrò et al., 2021*). In other words, the overall perception of the experience is affected by the artefacts of a log curve, segregation effect included (see *Figure 1*). Being produced instantaneously, pheromone deposition is likely linked to this information stream. However, when registering distances in memory ants need to be extremely precise. They possess an internal step counter and a visual odometer, which allows them to judge distances and return successfully to the nest (*Narendra, 2007*;

*Wittlinger et al., 2006*). Encoding steps logarithmically would be absolutely insufficient for accurate homing, given the need to already cope with errors introduced in the path integration process (*Merkle et al., 2006*; *Merkle and Wehner, 2010*; *Müller and Wehner, 1988*; *Schwarz et al., 2011*). For this, a linear, 1:1 correspondence is required. It is possible that distance is memorized linearly and in a separated stream from the value perception. When asked to choose, the ant can only compare the two memories of distances, which due to their linear nature are immune to the segregation effect (*Figure 6*). If the ant odometer truly is a rare example of linear perception, it would be an invaluable system for investigating information processing in insects, as it allows the roles of perception and post-perceptual processing to be disentangled.

The dissociation between immediate reaction and subsequent choice could also be explained by a separation between 'liking' and 'wanting'. The duality between 'liking' and 'wanting', first described in rats (*Berridge et al., 1989*) and then confirmed extensively in humans (*Brauer and De Wit, 1997*; *Leyton et al., 2002*; *Pool et al., 2015*), distinguishes two aspects of reward. 'Liking' refers to acutely perceived hedonic reactions, while 'wanting' refers to motivation and desire for a goal. While 'liking' and 'wanting' usually co-vary, these are neurologically separate processes, and can, in vertebrates, be separately inactivated (*Berridge et al., 1989*) or enhanced (*Berridge and Valenstein, 1991*; *Leyton et al., 2002*; *Treit and Berridge, 1990*). A dedicated network for 'wanting' has been recently discovered in insects as well (*Garcia and Dyer, 2022*; *Huang et al., 2022*), making it very likely to be present in *L. niger* ants as well. Under this perspective, the immediate reaction to experiencing the reward (i.e. the pheromone deposited) may correspond to how much the animal 'liked' each option. The choice of which reward to head for in the Y-maze task may reflect how much 'wants' either option. However, neurological evidence would be required to support this speculation. Moreover, further

experiments should test whether pheromone deposition and binary choice are indeed appropriate proxies of 'liking' and 'wanting', respectively.

Our work also demonstrates empirically that individual ants prefer near to distant food sources. Such a preference have been previously observed in nature (*Frank and Linsenmair, 2017*; *Nyamukondiwa and Addison, 2014*) but never directly tested. Ants also recruited significantly more to closer food source, as reported in this and other ant species as well (*Devigne and Detrain, 2006*; *Fewell et al., 1992*).

This study adds the bundling and segregation effect to the collection of value-distorting effects from behavioural economics which also affect animal value perception. Critically, by finding this effect in an invertebrate, we demonstrate that the complexities of the vertebrate brain are not required for these effects to manifest. Animal behaviour, much like human behaviour, can often be modelled as a value-maximizing system. However, deviations for strict economic rationality, such as the one demonstrated here, may play important roles in the animals' ecology, especially in their foraging behaviour and biotic interactions.

## Acknowledgements

TJC was supported by a Heisenberg fellowship from the Deutsche Forschungsgemeinschaft (CZ 237/4-1). MDA was funded by University of Regensburg 'Anreizsystem' funding to TJC.

## Additional information

### Funding

| Funder | Grant reference number | Author |
|---|---|---|
| Deutsche Forschungsgemeinschaft | CZ234/4-1 | Tomer J Czaczkes |
| Universität Regensburg | | Massimo De Agrò |

The funders had no role in study design, data collection and interpretation, or the decision to submit the work for publication.

### Author contributions

Massimo De Agrò, Conceptualization, Data curation, Software, Formal analysis, Supervision, Visualization, Methodology, Writing – original draft, Writing – review and editing; Chiara Matschunas, Investigation, Methodology, Writing – review and editing; Tomer J Czaczkes, Conceptualization, Supervision, Funding acquisition, Investigation, Writing – original draft, Project administration, Writing – review and editing

### Author ORCIDs
Massimo De Agrò ⓘ http://orcid.org/0000-0001-9284-5964

### Decision letter and Author response
Decision letter https://doi.org/10.7554/eLife.79314.sa1
Author response https://doi.org/10.7554/eLife.79314.sa2

## Additional files

### Supplementary files
• Supplementary file 1. Full analysis script. Entire statistical analysis code produced with R and Python, including data handling, figure code, and analysis results.

• MDAR checklist

• Source data 1. Raw data of the experiments. Raw data for the pilot and the three experiments. Metadata describing the columns of each csv file is included as a markdown.

**Data availability**

Raw data collected in the experiments and used for the analysis are available in Source data 1.

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
