## [Editor Report]

This innovative study investigates the presence of biases in value perception in ants. The authors were able to show that the distribution of rewards and costs influences perceived reward value in ants, an effect that was observed for pheromone deposition but not choice behaviour.

---

## [Decision Letter]

**Decision letter after peer review:**

Thank you for submitting your article "Bundling and segregation affects "liking", but not "wanting", in an insect" for consideration by *eLife*. Your article has been reviewed by 4 peer reviewers, and the evaluation has been overseen by a Reviewing Editor and Christian Rutz as the Senior Editor. The following individuals involved in the review of your submission have agreed to reveal their identity: Giorgia Silani (Reviewer #1); Erik Frank (Reviewer #2); Yoann Stussi (Reviewer #4).

Essential revisions:

Overall, the reviewers agreed that the manuscript is of broad interest for many behavioral scientists in the fields of behavioural economics, decision-making, and reward processing. We were also positively impressed by the open practices in openly sharing data and analysis code in a very clear and detailed manner. However, the manuscript requires substantial revisions of the interpretation and discussion of the data as well as supplementary data to rule out some important alternative explanations. While there seems to be a consensus concerning the value of the demonstrated bundling versus segregation effects, we also agree that the interpretation in terms of "wanting versus liking" appears too speculative and questionable given the data. Therefore, we strongly suggest removing the interpretation in terms of motivational and hedonic processes and staying closer to the actual bundling effects which are innovative by themselves. In terms of alternative interpretations of the data, there are some major confounds that we would like to see addressed with supplementary data or supplementary analysis.

1) Concerning the results from the Y maze behavior: For interpretation of the results of the Y maze in terms of choice behaviour it is necessary to demonstrate that the associations between the reward conditions and the runway scents were successfully learned. The pilot experiment differs in many ways from the main experiment, therefore one cannot rule out that the lack of preference in the Y maze simply reflects a lack of learning. Also related to the interpretation of the behavior in the Y maze, the choice test is administered under extinction, an analysis of the first choice only would allow to rule out that the absence of preference reflects extinction processes.

2) Concerning the results from the pheromone deposit: A possible alternative interpretation of the pheromone deposit measure would be that it simply reflects the caloric intake rather than food value, it is worth controlling that across the different reward conditions the total amount of food consumed does not systematically vary and that the pheromone release did not vary according to the quantity of food that was consumed. More generally, the manuscript needs to provide more information about how pheromone deposition was measured and on the specificities of this measure, such as its physiological bases, timing properties, and granularity. Importantly, the results from the pheromone deposit measure are interpreted as if they support the hypothesis that segregated rewards with bundled costs should be the most "liked" option relative to bundled rewards and costs and segregated rewards and costs. But the data reported does not fully support this: The difference between the 'segregated rewards' condition and the 'bundled' condition is not statistically significant when all ants are considered. More nuance is needed so that this statistically nonsignificant result does not appear as being treated as statistically significant.

3) Concerning main experimental manipulation: the experimental design relies on the assumption that ants can perceive drops spaced by 5 mm as being segregated (two of 0.2 µl and 1 ad libitum). Since this is the key manipulation on which the experimental manipulation relies, we would like to see data demonstrating that spacing two food drops by 0.5 mm induces a segregated perception in ants.

4) Concerning the statistical approach: The choice of the random-effects structure for the GLMM analyses should be clearly motivated and justified. In particular, random intercepts were modelled (despite repeated measurements) and an appropriate specification of the random-effects structure is pivotal to reaching correct inferences when using mixed-effects models.

*Reviewer #4 (Recommendations for the authors):*

1) Concerning the ethics statement, it is mentioned 'No' next to animal subjects. Does this mean that no ethical approval was required or submitted for this study? If so, is this a typical procedure for studying with ants?

2) Across the paper, there is a lot of emphasis that the study is the first to report a bundling effect and a "wanting" / "liking" dissociation in ants. However, I don't think such an emphasis is necessary or even warranted as it doesn't provide helpful information to evaluate the importance or robustness of the results. I feel that the manuscript would benefit from removing these mentions or at least keeping them to a minimum.

3) It would be beneficial to report how the sample size for the main experiment was established or provide some considerations about the power of the study (e.g., a power sensitivity analysis to indicate the smallest effect size in terms of the odds ratio that could be detected with a certain power threshold given the sample size).

4) The degrees of freedom of the t-tests associated with the effects from the generalised linear mixed models (GLMM) should be consistently reported.

5) The choice of the random-effects structure for the GLMM analyses should be clearly motivated and justified. This is particularly important because only random intercepts were modelled (despite repeated measurements) and an appropriate specification of the random-effects structure is pivotal to reaching correct inferences when using mixed-effects models (see Barr et al., 2013. https://doi.org/10.1016/j.jml.2012.11.001; Bates et al., 2015. https://arxiv.org/abs/1506.04967; Frossard & Renaud, 2019. https://arxiv.org/abs/1903.10766).

6) Complementing the results with confidence intervals (e.g., 95% for statistical tests reporting a t statistics) around the effect sizes of interest could provide valuable information and be a nice addition to the manuscript. A useful R package to calculate confidence intervals for effect sizes is effectsize (https://cran.r-project.org/web/packages/effectsize/index.html; https://easystats.github.io/effectsize/).

7) I would like to commend the authors for openly sharing their data and analysis code in a clear and detailed manner.

---

## [Author Response]

Essential revisions:Overall, the reviewers agreed that the manuscript is of broad interest for many behavioral scientists in the fields of behavioural economics, decision-making, and reward processing. We were also positively impressed by the open practices in openly sharing data and analysis code in a very clear and detailed manner. However, the manuscript requires substantial revisions of the interpretation and discussion of the data as well as supplementary data to rule out some important alternative explanations. While there seems to be a consensus concerning the value of the demonstrated bundling versus segregation effects, we also agree that the interpretation in terms of "wanting versus liking" appears too speculative and questionable given the data. Therefore, we strongly suggest removing the interpretation in terms of motivational and hedonic processes and staying closer to the actual bundling effects which are innovative by themselves. In terms of alternative interpretations of the data, there are some major confounds that we would like to see addressed with supplementary data or supplementary analysis.1) Concerning the results from the Y maze behavior: For interpretation of the results of the Y maze in terms of choice behaviour it is necessary to demonstrate that the associations between the reward conditions and the runway scents were successfully learned. The pilot experiment differs in many ways from the main experiment, therefore one cannot rule out that the lack of preference in the Y maze simply reflects a lack of learning. Also related to the interpretation of the behavior in the Y maze, the choice test is administered under extinction, an analysis of the first choice only would allow to rule out that the absence of preference reflects extinction processes.2) Concerning the results from the pheromone deposit: A possible alternative interpretation of the pheromone deposit measure would be that it simply reflects the caloric intake rather than food value, it is worth controlling that across the different reward conditions the total amount of food consumed does not systematically vary and that the pheromone release did not vary according to the quantity of food that was consumed. More generally, the manuscript needs to provide more information about how pheromone deposition was measured and on the specificities of this measure, such as its physiological bases, timing properties, and granularity. Importantly, the results from the pheromone deposit measure are interpreted as if they support the hypothesis that segregated rewards with bundled costs should be the most "liked" option relative to bundled rewards and costs and segregated rewards and costs. But the data reported does not fully support this: The difference between the 'segregated rewards' condition and the 'bundled' condition is not statistically significant when all ants are considered. More nuance is needed so that this statistically nonsignificant result does not appear as being treated as statistically significant.3) Concerning main experimental manipulation: the experimental design relies on the assumption that ants can perceive drops spaced by 5 mm as being segregated (two of 0.2 µl and 1 ad libitum). Since this is the key manipulation on which the experimental manipulation relies, we would like to see data demonstrating that spacing two food drops by 0.5 mm induces a segregated perception in ants.4) Concerning the statistical approach: The choice of the random-effects structure for the GLMM analyses should be clearly motivated and justified. In particular, random intercepts were modelled (despite repeated measurements) and an appropriate specification of the random-effects structure is pivotal to reaching correct inferences when using mixed-effects models.

We note that many of the comments were shared by multiple reviewers, and the editor effectively summarized the main, shared positions in the points below. To make our response easy to navigate, we added a numbering system to the different points made. We will include our main arguments as answers to the editors comments, and we will refer to those, while adding further specific comments, in the answers to reviewers below when these points are raised.

1. The liking vs wanting frameworkWhile there seems to be a consensus concerning the value of the demonstrated bundling versus segregation effects, we also agree that the interpretation in terms of "wanting versus liking" appears too speculative and questionable given the data. Therefore, we strongly suggest removing the interpretation in terms of motivational and hedonic processes and staying closer to the actual bundling effects which are innovative by themselves.

Having seen how this is a shared opinion of multiple reviewers, we agree with greatly reducing our claims about liking vs wanting as an explanation to our results. Indeed, the experiments were not designed to test this specific segregation, and as such cannot make broad claims on its presence in our experimental subjects.

However, we still believe we can’t ignore the unusual split result between the binomial choice test and the pheromone deposition. This was unexpected for us, as in virtually all studies we ever carried out using this odor learning procedure we found a clear response in the Y-maze choice tests. Occasionally this was not mirrored in the pheromone data, but never the other way around. Note that it was never our intention to suggest that our paper was designed to test liking vs wanting. Indeed in the introduction we are very clear in saying that this reasoning came after observing the results (L99-106).

We however agree that our inclusion of this argument was too wide for this manuscript, given the ideas’ speculative nature. We agree that this paper does not demonstrate the presence of a separation in liking vs wanting in ants, but we feel we should at least mention it as a possible explanation to the difference in the effect between pheromone deposition and binomial choice. This position we believe is also sustained by the recently published paper by Huang, J. et al. (2022), that came out just a month after out submission to *eLife*.

Accordingly, we changed the title of the paper to “Bundling and segregation affects pheromone deposition, but not choice, in an ant” and removed every mention of the liking vs wanting literature from the introduction, focusing solely on economic theories and bundling vs segregation. In the discussion, we reduced greatly its contribution, but we mention liking vs wanting as a possible, yet speculative, explanation to our observation. We limit this discussion to one paragraph and refer to the recent Huang et al. ^(2022)^ paper as a demonstration of the existence of this dichotomy in insects, and our observation as a possible effect. We also make very clear that this is speculation, note statements of fact, by ending the paragraph with “However, neurological evidence would be required to support this speculation.” The mismatch between memory and perception is maintained as an alternative explanation, indicating also how this mismatch may cause a failure in memory itself, as was suggested by some reviewers.

In terms of alternative interpretations of the data, there are some major confounds that we would like to see addressed with supplementary data or supplementary analysis.

Regarding some of the requested additional data, intended to exclude alternative explanations, we believe we have sufficient evidence, stemming either from the literature or from other pilot experiments performed by us. We will present our argument below when brought up. We are of course open to discussing collecting more data if our position was to be found unsatisfactory.

2. The Y-maze results2.1. Presence of learningConcerning the results from the Y maze behavior: For interpretation of the results of the Y maze in terms of choice behaviour it is necessary to demonstrate that the associations between the reward conditions and the runway scents were successfully learned. The pilot experiment differs in many ways from the main experiment, therefore one cannot rule out that the lack of preference in the Y maze simply reflects a lack of learning.

Yes, this is an issue which affects all ‘negative results’ in behavior-based learning assays: it is impossible to distinguish lack of learning from learning but without a resultant behavioural response. Thus, as far as we can tell, is impossible to demonstrate learning under this exact bundling vs segregation setup as separate from preference.

However, we have good reasons to expect learning to have taken place, given their successful learning in other contexts using an identical odour-association choice assay. These ants learn such associations very well in a multi-reward context. In 2018 we performed a control experiment required for a different experiment. In that experiment, as in the experiment we report here, ants had to encounter multiple sucrose solution drops over the course of one visit, and associate all of those to an odor. We designed the pilot to demonstrate the ability of ants to associate a smell with a multi-reward visit. Each ant (n=20) would be allowed onto a 20cm long scented runway. At the end of it, the animal would encounter a small 0.2ul drop of sucrose solution. The solution could be either 1.5M or 0.25M, depending on the condition. After the ant had fully consumed the drop, a second one would be presented, and after that a third one, that the ant consumed until satiated. All the three drops had the same molarity. The ant was then allowed back to the nest, unloaded, and allowed to make a second visit. Here the ant would experience a differently scented runway, and presented with a different molarity drop (0.25M if 1.5M for the first visit and vice-versa). This procedure was repeated 8 times, for a total of 4 visits for each odor. In essence, this experiment is almost identical to the “Segregated reward” condition as presented in this manuscript, where we also presented a succession of three identical drops. At test, 90% of the ants chose the scent associated with high molarity for the first choice, 75% did so for the second choice, 70% for the third. This experiment demonstrates the ability of ants to associate an odor with a multi-reward experience. We are attaching the raw data for this pilot experiment to the response. If the reviewers and editor were to find this data crucial to the discussion of the current experiment, we can add the full pilot as a supplement.

Regardless, in the new version of the manuscript we give more space to the perception vs memory explanation (Figure 6) as an alternative explanation to the data. We raise the possibility that the ants simply didn’t learn the task, but the bundling vs segregation effect had an effect or weighting on their perception. We find this explanation unlikely, given the corroborating evidence we presented, and thus lean more towards an equal weight in memory but not in perception.

2.2. Results under extinctionAlso related to the interpretation of the behavior in the Y maze, the choice test is administered under extinction, an analysis of the first choice only would allow to rule out that the absence of preference reflects extinction processes.

Regarding the test being administered under extinction, note that visit number was always included in the models in order to observe whether the subsequent testing had a decreasing effect of choice. Even when no difference was found between subsequent testing visits, we included the relevant post-hoc analyses too, regardless of the preference in the very first testing visit.

When looking only at the first choice, for condition 1, the ants chose the segregated option 51.6% of the times (p=1). For condition 2, they chose the segregated reward option 49.7% of the times (p=1). Lastly, in condition 3 they chose the segregated option 34.7% of the times (p=1). This data is available in the Supplementary file 1 (before supplement ESM2).

3. Pheromone deposition3.1. The origin of pheromone depositedConcerning the results from the pheromone deposit: A possible alternative interpretation of the pheromone deposit measure would be that it simply reflects the caloric intake rather than food value, it is worth controlling that across the different reward conditions the total amount of food consumed does not systematically vary and that the pheromone release did not vary according to the quantity of food that was consumed. More generally, the manuscript needs to provide more information about how pheromone deposition was measured and on the specificities of this measure, such as its physiological bases, timing properties, and granularity.

We feel fairly confident in excluding the possibility that pheromone deposited is linked to caloric consumption. First of all, ants participating in this experiment do not consume the liquid. The reward gets stored in the social stomach and not assimilated, and gets then passed on other members of the colony. As such, metabolic processes should not take place during the experiment.

More importantly, however, good evidence exists that, in this species, pheromone deposition is released after a specific volume threshold is consumed (A. C. Mailleux et al., 2000; A.-C. Mailleux et al., 2003, 2005). Critically, after passing this limit, ants which consume the same quality of food in the same manner deposit the same amount of pheromone, regardless of the specific volume drunk. Ants will not freely return to the nest before reaching this threshold, if allowed to feed further, and the gaster only becomes very visibly distended after this threshold. Thus, we are confident that all the ants in our experiment consumed more than this threshold food volume, and so even if the different treatments cause small changes in volume consumption, this should not affect their pheromone deposition.

Finally, regardless of the reported literature, we feel that even if the ants were to deposit more pheromone only as a byproduct of different caloric intake, the results would still support our interpretation. The only difference between the treatments is the segregation of rewards and costs itself. All the drops presented are of the same sizes across treatments, and they all carry the same molarity. As such, if the ants were to consume the food at a different rate or in a different amount, this variance would have to be caused by the segregation effect itself. Some evidence is available demonstrating that ants tend consume more overall volume for moderately high sucrose concentrations ^(Sola & Josens, 2016)^, which would indicate that the segregation has an effect on the perceived value. In other words, even if pheromone was linked to caloric consumption, the deposition would act as a proxy of consumption rate, just another modulated measure.

We also now provide, when first introducing the pheromone deposition measurement, some more detail on the behavior, and how it correlates with the (perceived) value of a resource.

3.2. The presence of not statistically significant resultsImportantly, the results from the pheromone deposit measure are interpreted as if they support the hypothesis that segregated rewards with bundled costs should be the most "liked" option relative to bundled rewards and costs and segregated rewards and costs. But the data reported does not fully support this: The difference between the 'segregated rewards' condition and the 'bundled' condition is not statistically significant when all ants are considered. More nuance is needed so that this statistically nonsignificant result does not appear as being treated as statistically significant.

Apologies, we were not very clear in our discussion here, and didn’t sufficiently distinguish out original hypotheses from our interpretation of the results. Our interpretation was that Bundling vs Segregation has an effect only on costs, but not on gains (L428-435, original manuscript version). This idea is derived from the data as is, considering the Bundling vs Segregated reward as identical, the bundling as preferred to the segregated all, the segregated reward being preferred to segregated all. Under this interpretation, the segregated rewards condition should NOT result different from the bundled condition, as in both the costs are bundled, and they only differ in the gains (L435-439, original manuscript version). For the same reason, Segregated reward (that presents bundled cost) is preferred to segregated all (that presents segregated costs); bundled (bundled cost) is preferred to segregated all (segregated costs).

We understand that the confusion originates from us repeating our a priori hypothesis in the discussion, which instead expected to see a difference in segregated reward vs bundles. We feel that is more honest of us to declare our a priori hypothesis, and then specify what the data suggest, rather than to give the impression to having had the second hypothesis all along. We address this issue by explicitly distinguishing our a priori hypothesis from our interpretation. L391, L394, L398, L403-404

Another reason for confusion is that we discussed at length the nuance of the results. We felt that declaring a result to be definitely at chance level when the p-value is 0.059 would be careless, especially considering the pattern of preference when the initial encounter is included.

We have now reworded the discussion and made clearer the expectation for each condition depending on the working hypothesis.

4. The segregation effectConcerning main experimental manipulation: the experimental design relies on the assumption that ants can perceive drops spaced by 5 mm as being segregated (two of 0.2 µl and 1 ad libitum). Since this is the key manipulation on which the experimental manipulation relies, we would like to see data demonstrating that spacing two food drops by 0.5 mm induces a segregated perception in ants.

This issue is related to point 2 – how can we distinguish a lack of response to a perceived stimulus from a stimulus that is not perceived? We cannot. Indeed, we were very careful not to make strong claims that ants do not respond to reward segregation. We now state explicitly “…either because the ants do not segregate gains, or because our manipulation failed to trigger segregation”.

Happily, for our main claim that bundling and segregation influences the behavior of ants, it does not matter: this treatment was added in case reward segregation affected the ants in an equal and opposite way to cost segregation, since they both co-occur in the “all segregated” treatment. Has the response to “all segregated” and “all bundled” been indistinguishable, we would have required the “reward segregated” treatment to interpret the result: a lower reward perception of “segregated all” compared to “segregated all” would have been taken as evidence that reward and cost segregation, when co-occurring, balance each other out. As it happens, however, there was a difference between “all segregated” and “all bundled”, showing that costs are experienced differently if bundled or segregated. Unclear is only whether rewards are perceived as segregated, but have a weaker effect that segregated costs, or whether the ants only perceived the costs, but not the rewards, as segregated.

In summary: we agree completely that we cannot know whether the reward segregation treatment was really perceived as segregated by the ants. However, given the results, it does not matter for our main message.

To make this clear, we have now added these considerations to the discussion. L410-416

5. The choice of random effect structureConcerning the statistical approach: The choice of the random-effects structure for the GLMM analyses should be clearly motivated and justified. In particular, random intercepts were modelled (despite repeated measurements) and an appropriate specification of the random-effects structure is pivotal to reaching correct inferences when using mixed-effects models.

Our choice of random effect structure was informed by our knowledge of the study species. Different colonies can have an overall different tendency to deposit pheromone. In each colony, different individuals can have a different tendency to deposit overall more or less pheromone. For this reason, the individual ants nested in colonies are included as a random intercept in the model: the overall pheromone deposited may vary per each individual independently from the treatment. However, we do not have the same expectation for across trials changes: an over-performer ant will consistently over-perform in terms of pheromone deposited, with no expectation for it to differentially change behaviour (Beckers et al., 1992). However, the evidence is sparse, and we are mostly relying on our own experience. Given this, prompted by one reviewer comments, we tried including visit number as a random slope for models looking at pheromone deposition (Binomial choice models only presented 3 subsequent visits, all being of equal treatment. As such we only included a random intercept). This turned out to be not feasible all the time, causing a failure in model convergence. We believe that the reason is the low number of repetitions per condition per subject: each performs only 4 visits to each segregation option. Moreover, visit 1 almost always results in no pheromone deposition, due to the existence of pre-training and generally low deposition on the first 1-2 visits to a food source, leaving probably too few repetitions to estimate the random slope. Following the suggestions of the reviewer, we tried simplifying the random effect structure in those cases, including Visit number as random intercept. This sometimes fixed the convergence failure, in which case we maintained it in the model, while in others the model kept failing to converge. In those cases we were forced to remove it from the final model.

Reviewer #4 (Recommendations for the authors):1) Concerning the ethics statement, it is mentioned 'No' next to animal subjects. Does this mean that no ethical approval was required or submitted for this study? If so, is this a typical procedure for studying with ants?

Indeed, ants and all invertebrates (notably excluding cephalopods, and recently some crustaceans) are not protected by European law (or by any other country as far as we know) in research. As such, no ethical approval is required to perform experiments on these animals.

2) Across the paper, there is a lot of emphasis that the study is the first to report a bundling effect and a "wanting" / "liking" dissociation in ants. However, I don't think such an emphasis is necessary or even warranted as it doesn't provide helpful information to evaluate the importance or robustness of the results. I feel that the manuscript would benefit from removing these mentions or at least keeping them to a minimum.

We agree with the reviewer. See point 1.

3) It would be beneficial to report how the sample size for the main experiment was established or provide some considerations about the power of the study (e.g., a power sensitivity analysis to indicate the smallest effect size in terms of the odds ratio that could be detected with a certain power threshold given the sample size).

Sample size was set *a-priori*, but based on pragmatic considerations: we knew how much research time we could allocate to data collection, and could make a good estimate of how many ants could be tested in this time. This was then used to set the sample size. Using power analyses can be tricky when we have poor *a-priori* estimates for the effect size. We consider our pragmatic a priori sample size setting approach to be functional while avoid the possibility of p-hacking.

4) The degrees of freedom of the t-tests associated with the effects from the generalised linear mixed models (GLMM) should be consistently reported.

DF are now reported in the text, thank you.

5) The choice of the random-effects structure for the GLMM analyses should be clearly motivated and justified. This is particularly important because only random intercepts were modelled (despite repeated measurements) and an appropriate specification of the random-effects structure is pivotal to reaching correct inferences when using mixed-effects models (see Barr et al., 2013. https://doi.org/10.1016/j.jml.2012.11.001; Bates et al., 2015. https://arxiv.org/abs/1506.04967; Frossard & Renaud, 2019. https://arxiv.org/abs/1903.10766).

We would like to thank the reviewer greatly for the clarification provided via email. It helped us immensely in navigating the development of a random effect structure. We answered fully to the comment in the general response, and the changes are now appreciable in the supplement.

6) Complementing the results with confidence intervals (e.g., 95% for statistical tests reporting a t statistics) around the effect sizes of interest could provide valuable information and be a nice addition to the manuscript. A useful R package to calculate confidence intervals for effect sizes is effectsize (https://cran.r-project.org/web/packages/effectsize/index.html; https://easystats.github.io/effectsize/).

Thank you for suggesting the package! We did not know it and we found it quite useful. We added a calculation of the effect size in the analysis provided in the supplement. We however did not add it in the main text, to maintain readability. Indeed, effect size calculated by the effectsize package is directly derived from DF and t value, and as such we felt it would be redundant.

7) I would like to commend the authors for openly sharing their data and analysis code in a clear and detailed manner.

Many thanks! Open science is very important to us, and we work hard on it. It is pleasing to see that it is sometimes noticed.